# Serum Tumor Markers for Muscle-Invasive Bladder Cancer in Clinical Practice: A Narrative Review

**DOI:** 10.3390/cancers17050728

**Published:** 2025-02-21

**Authors:** Chirag Doshi, Mazyar Zahir, Anosh Dadabhoy, Domenique Escobar, Leilei Xia, Siamak Daneshmand

**Affiliations:** Catherine and Joseph Aresty Department of Urology, University of Southern California, 1441 Eastlake Ave. NOR 7416, Los Angeles, CA 90089-9178, USA; chirag.doshi@med.usc.edu (C.D.); mazyar.zahir@med.usc.edu (M.Z.);

**Keywords:** urothelial carcinoma, serum tumor markers, muscle-invasive bladder cancer

## Abstract

Bladder cancer is a significant health concern that often progresses without early symptoms, complicating timely detection and treatment. This review intends to summarize the application of serum tumor markers (STMs)—proteins in the blood that can signal the presence of cancer—as a diagnostic and monitoring tool for patients with muscle-invasive bladder cancer (MIBC), supplemented with real-life cases from our institute. We demonstrate how STMs can aid in prognosis measurement, assess treatment effectiveness, and identify recurrence earlier, thereby potentially improving patient outcomes. By presenting our findings and clinical insight, we hope to highlight the practical advantages of incorporating STMs into MIBC management, which could enhance patient care and stimulate further research in the medical community.

## 1. Introduction

Bladder cancer (BC) is one of the most burdensome cancers worldwide, responsible for over 200,000 deaths in 2020. Most of these deaths are from muscle-invasive bladder cancer (MIBC) [1,2]. Despite comprehensive treatment regimens comprising neoadjuvant chemotherapy, radical cystectomy (RC) and adjuvant immunotherapies, a large group of MIBC patients continue to progress to more advanced stages without any early clinical manifestations.

For decades, after radical cystectomy for MIBC, the standard for monitoring for the risk of MIBC has been cross-sectional imaging. When diagnosed with MIBC, patients require frequent cross-sectional imaging to assess for evidence of metastases, progression or recurrence after treatment [3,4]. However, microscopic recurrences may be missed in cross-sectional imaging. Moreover, very early recurrences can be missed in the time that the patient is waiting for their next follow-up. In recent years, diagnostic kits detecting circulating tumor DNA (ctDNA) have gained considerable attention [5]. However, their high cost, limited accessibility in remote and under-resourced medical settings, and the requirement for tissue specimens hinders their wide adoption.

Considering the above-mentioned points, a non-invasive, affordable and accessible tool that expedites the detection of MIBC recurrence could significantly improve disease management and prevent progression. We and others have demonstrated that three epithelial STMs, i.e., Carbohydrate Antigen 19-9 (CA 19-9), Cancer Antigen-125 (CA-125) and carcinoembryonic antigen (CEA), can serve as effective prognostic tools in MIBC patients [6]. For the estimated 31–70% of patients who have at least one abnormal STM prior to cystectomy, there remains a potential benefit for STMs being used for assessing prognosis [7,8]. These STMs have been evaluated as representative biomarkers for various malignancies, highlighting their potential in cancer assessment and monitoring.

CA-125, also known as mucin 16, is a large surface glycoprotein, known to be involved in the modulation of epidermal growth factor receptor (EGFR) phosphorylation, and is commonly used for the diagnosis and management of ovarian cancer [9]. Considering the established role of the EGFR signaling pathway in BC pathogenesis, CA-125 can be a suitable candidate for MIBC surveillance [10]. Similarly, CA 19-9, which is another surface glycoprotein, has been shown to be elevated in patients with gastrointestinal malignancies, particularly pancreatic cancer. High serum levels of CA 19-9 have been reported in patients with metastatic BC [11]. CEA is one of the most used tumor markers in oncology. Previous studies showed that CEA levels increased in about a quarter of patients with advanced BC. Additionally, CEA levels appear to have an inverse correlation with clinical response [12].

Although small studies have reported the value of these STMs in BC, their potential roles as assessment tools for BC have been relatively understudied, with limited clinical research conducted on their application. Consequently, we aimed to provide a detailed description of the current applications and potentials of STMs in MIBC patients and share some of our successful experiences with STMs in the clinical setting.

## 2. STMs for Prognosis Assessment

Some STMs enable physicians to assess the likelihood of specific disease outcomes by stratifying patients into different risk groups, while also assisting with treatment personalization. In some malignancies, such as breast cancer, STMs may be used as markers of treatment toxicity prediction. In BC, higher levels of CA 19-9 are directly associated with higher pathologic stages, characterized by muscular layer invasion and metastasis ultimately leading to poorer survival outcomes [13,14]. Accumulating data suggests that CA19-9 levels may reflect tumor burden and aggressiveness in BC patients [11,15,16]. However, to date, there has been no validated prognostic marker for BC.

Our institution was among the first to evaluate the potential role of STMs in BC starting in 2004. In the first study published from our institution in 2014, the prognostic value of CA-125 and CA 19-9 was evaluated. The results were promising, revealing a direct association between CA-125 levels and both extravesical extension and lymph node metastasis. Moreover, patients who had increased levels of CA-125 or CA 19-9 were shown to have worse overall survival [17]. Five years later and in our second study, we demonstrated that elevated precystectomy levels of CA 19-9 and CEA were independent predictors of worse 3-year overall survival, with 2.7- and 2-fold increased risk of death, respectively. Moreover, elevated CA 19-9 level was shown to be an independent predictor of an approximately 2.8-fold increase in recurrence risk at the 3-year follow-up [7]. Findings from our last effort in 2019 indicated that persistently elevated STMs after neoadjuvant chemotherapy were associated with pathologic upstaging and worse survival outcomes [6]. A recent study by Yuk et al. also investigated the same serum tumor markers including β-hCG and concurred that patients with elevated CA-125 had a dramatically worse survival (HR: 6.21 95% CI:1.34–32.16) [18]. Their studies also confirmed that combining tumor markers levels improved the ability to assess prognosis more than one marker alone.

## 3. STMs for Evaluation of Therapy Response

Contrary to their limited role in prognostication and diagnosis, STMs have a relatively established role as indicators of therapeutic response. In a study by Izes et al., CA-125 was shown to be a strong predictor of tumoral activity and treatment response in patients with advanced BC [8]. According to their findings, 16 out of 30 (53%) patients with disease progression had simultaneous increases in their CA-125 levels. Notably, in 5 cases where the clinical course suggested treatment failure and deterioration, increases in CA-125 levels were the only indicator, as no clear evidence of disease progression was observed on imaging. Moreover, CA-125 levels decreased by 42% on average after chemotherapy in their cohort, further suggesting its proposed role as a treatment response marker. It must be noted that 71% of their cohort had increased levels of CA-125 initially, suggesting that this STM may be a feasible marker for treatment response in these patients.

Interestingly, Yaegashi et al. reported that elevated levels of CA 19-9 in patients with metastatic BC were associated with a better response to chemotherapy [16]. This appears to contrast with the findings of Ahmadi et al., which indicated worse survival outcomes with higher CA 19-9 levels prior to cystectomy [7]. However, it is important to note that, unlike the latter study, Yaegashi et al. focused on metastatic disease. It is plausible that once the tumoral cells reach the metastatic stage, they become more undifferentiated and, despite producing higher levels of tumor markers, may respond better to chemotherapy.

Parallel to previous studies, Washino et al. demonstrated the feasibility of STMs in evaluating therapeutic response [19]. Their results showed a strong association between decreases in CA 19-9 and CA-125 levels and therapeutic response, particularly in the neoadjuvant chemotherapy setting. Notably, patients experienced a greater than 50% reduction in their STMs following chemotherapy compared to baseline levels. Conclusively, there seems to be consensus across studies that, as Cook and colleagues suggested more than two decades ago, clinical response and tumor marker responses are strongly correlated, provided the patient has at least one elevated STM prior to treatment. To improve detection rates, Cook et al. recommended utilizing a panel of STMs instead of a single STM to increase the likelihood of identifying at least one increased STM and utilize it to monitor clinical response [20].

## 4. STMs for Bladder Cancer Surveillance: Our Experience

There is limited information on STMs for surveillance and early detection of recurrence in BC patients. Over the last 12 years, we have actively monitored STMs in addition to the standard follow-up protocol at our institution. Our experience suggests that STM elevations can indicate recurrence earlier than imaging or clinical symptoms of progression. Herein, we present three representative cases in which STMs played a crucial role in guiding clinical management by either revealing early recurrence or showing that the ongoing therapy was ineffective. Full description of all cases can be found in Appendix A.

Our first case is a 55-year-old woman with metastatic MIBC who was referred to our clinic in late 2022. She underwent standard systemic chemotherapy with Gemcitabine and Cisplatin and her post-treatment evaluation with cystoscopy, PET scan, and STMs indicated a complete response. She was subsequently enrolled in a clinical trial for maintenance therapy with Avelumab. Her STMs remained stable until early 2024, when an increase in CA19-9 levels was noted while she was still being treated with maintenance Avelumab. Cystoscopy at that time showed a very small recurrent tumoral mass, which was deemed insignificant and did not warrant immediate treatment. Additionally, abdominopelvic and chest CT scans did not show any evidence of local or distant recurrence. Given the rising CA 19-9 levels, a restaging PET scan was obtained which revealed bilateral recurrent metastases in pelvic lymph nodes. Consequently, Avelumab was discontinued, and the patient was started on Pembrolizumab and Enfortumab Vedotin. Her CA 19-9 level dropped to less than half their previous value in less than a month. Her last lab evaluation in mid-2024 showed a further decrease in CA 19-9, indicating a favorable response to her new treatment (see Figure 1).

Our second patient was a 65-year-old man with high-grade papillary MIBC diagnosed in early 2020. He could not tolerate neoadjuvant Gemcitabine and Cisplatin and subsequently developed severe bilateral hydronephrosis. Consequently, he was referred to our urology clinic for upfront RC in mid-2020. Preoperative staging imaging was negative for metastasis but he had high CA 19-9 levels. However, surgery was aborted upon direct inspection of extensive peritoneal seeding of the tumor. His treatment plan was changed to dose-dense Methotrexate, Vinblastine, Doxorubicin, and Cisplatin (ddMVAC). Despite this treatment, CA 19-9 remained elevated, and CA-125 also showed a small increase. Upon completion of ddMVAC therapy, maintenance Avelumab was started for him. His STMs started rising in late 2020 and all three reached abnormally high levels despite receiving Avelumab. This increase prompted expedited imaging with PET/CT scan which suggested disease progression. Hence, his treatment regimen was changed to Enfortumab Vedotin, which managed to control and reduce CA 19-9 dramatically through the next 4–5 months. Unfortunately, CA 19-9 level rose significantly again after this and repeat imaging revealed progression of disease, leading to a switch to Sacituzumab Govitecan. Later, his STMs levels continued to fluctuate, forcing change in treatment plan from Sacituzumab Govetican to palliative RC in mid-2021 and later to salvage Gemcitabine in late 2021, Paclitaxel in early 2022, and eventually Carboplatin as the last resort. Unfortunately, he succumbed to cancer in mid-2022 (see Figure 2).

Our third patient was a 68-year-old woman who was first diagnosed with non-muscle invasive BC in mid-2017; however, this progressed to MIBC by 2021, manifesting as severe abdominal pain due to bilateral hydronephrosis. A positive biopsy and elevated levels of all three STMs, specifically extremely high levels of CA 19-9, were observed at the time of progression. She was started on Gemcitabine/Carboplatin/Atezolizumab, with all STMs decreasing tremendously in 2 months. However, post-chemotherapy imaging revealed a peritoneal nodule indicative of carcinomatosis. Treatment was hence switched to Gemcitabine/Cisplatin, followed by RC. STMs were normalized on the first postoperative appointment. She then started adjuvant Nivolumab. Despite this, her CA 19-9 levels started to rise in late 2021. PET and CT scans did not demonstrate any disease progression, but her CA 19-9 levels continued to increase until mid-2022. After discussion, her treatment was changed to Enfortumab Vedotin which significantly decreased her CA 19-9 level. However, by late 2022, her CA 19-9 dramatically increased again. At that time, an instance of significant abdominal pain prompted an emergent exploratory laparotomy, revealing a large bowel obstruction due to a retroperitoneal tumor and carcinomatosis. She was later treated with Cisplatin and Gemcitabine with decreasing STMs into mid-2023, but unfortunately, she later succumbed to her disease (see Figure 3).

## 5. Discussion

The three cases discussed above highlight a sample of our experience with STMs. Even in the second and third cases where the disease was aggressive, progressing, and ultimately fatal, the STMs prompted investigations and discussions that led to necessary treatment changes. Our first case represents our most dramatic example of the utility of STMs, as the patient’s survival can be at least partially attributed to their use. In this case, the rise in STMs prompted expedited PET imaging, which revealed BC recurrence which had not been detected with earlier CT scans. This finding led to the discontinuation of Avelumab and initiation of Pembrolizumab and Enfortumab Vedotin, ultimately resulting in complete remission of the disease. In the second case, elevated STMs prompted earlier imaging twice in their course. In both instances, disease progression was detected, which would have otherwise gone unnoticed until a subsequent follow-up visit, had it not been for the STM-driven expedited imaging. Despite the patient’s poor outcome, the initial change to Enfortumab Vedotin resulted in a dramatic response, likely extending survival by several months. Moreover, although the subsequent switch to Sacituzumab Govetican did not improve their condition, STMs still facilitated an earlier detection of disease. Similarly, in the third case, an abrupt increase in STMs prompted further evaluation. Despite no evidence of recurrence on imaging, a change in treatment led to a significant improvement in both the patient’s STMs profile and overall clinical condition. These cases underscore the critical role of STMs in monitoring disease activity and guiding treatment decisions.

Surveillance of MIBC patients for potential recurrence remains a significant challenge in clinical practice. While traditional diagnostic methods (i.e., imaging modalities) are effective, they have several limitations including costliness, radiation exposure, and reduced sensitivity for detecting early disease recurrence. To address these shortcomings, alternative diagnostic approaches have been proposed, including STMs, DNA methylation biomarkers, ctDNA, micro-RNAs and urinary extracellular vesicles (EVs) [21,22,23,24]. Among these, ctDNA is the most widely utilized for monitoring BC recurrence by offering real-time insights into tumor dynamics.

Previous studies have shown that ctDNA has acceptable sensitivity and specificity for detecting minimal residual disease and even predicting relapse in BC patients [25]. However, like all novel diagnostic tools, ctDNA has disadvantages including high costs, the need for specialized infrastructure and equipment, and the requirement of skilled technicians to accurately perform the test. These factors limit its accessibility, particularly in low- and middle-income countries [26]. Additionally, ctDNA relies on the DNA collected from the initial pathological specimen, which introduces several limitations. First, different areas within large tumors may have varying DNA signatures, making DNA sequencing from all these regions and varying signatures very challenging. Consequently, relapses originating from adjacent, unaccounted-for malignant tissue may go undetected. Small tumor masses have their own challenges, as attaining sufficient pathological specimen to establish a reliable tumor signature can be particularly difficult. Moreover, tumors are dynamic and undergo continuous genetic alterations, meaning that tissue obtained during an initial transurethral resection of the bladder tumor (TURBT) may not accurately reflect future genetic alterations in the relapsing malignancy [27]. This could significantly reduce the surveilling efficacy of ctDNA. This hurdle has even led to discussions about the potential need for “re-informing” the tumor signature after radical cystectomy for better sensitivity. While some assays may offer enhanced sensitivity in detecting ctDNA in a patient’s bloodstream, the increase in sensitivity often comes at the expense of decreased specificity. Another significant obstacle is that ctDNA can only detect cancer once it has progressed to a stage where it sheds enough DNA into the bloodstream to be detectable. Nonetheless, in many instances, tumoral cells do not undergo apoptosis or necrosis at a rate that leads to sufficient DNA shedding into the bloodstream [28].

In this context, and considering the limitations of the above-mentioned methods, STMs (i.e., CA-125, CA 19-9, and CEA), are a promising adjunct for regular monitoring due to their lower cost, non-invasiveness, and feasibility even in resource-limited settings. These STMs have shown acceptable validity, with ample research and evidence supporting their utilization for the 31–70% of patients that have an elevated STM value at the time of cystectomy. They can be particularly valuable in the interim period between two consecutive follow-ups. Typically, MIBC follow-up appointments are scheduled every 3 to 6 months during the first three years, with the aim of maximizing diagnostic efficacy while minimizing radiation exposure and financial burden [29]. However, patients often remain unmonitored in the interval period between these follow-up appointments. STMs can play a valuable role in these time intervals. A monthly STM panel could be used as a cost-effective and reliable diagnostic tool to follow patients during these intervals. In the moment an increase in STM levels was noticed, the medical team may decide to expedite imaging to detect any possible recurrence in a timely manner.

However, the potential of STMs must be considered against the backdrop of their limitations. First and foremost is the issue of specificity. Although previous studies have suggested that the CA-125, CA 19-9, and CEA panel can be sensitive for MIBC, their specificity is a major concern. Elevations in these STMs may occasionally indicate other malignancies—such as CA 19-9 and CA-125 indicating pancreatic and ovarian cancers, respectively—or even non-malignant conditions, such as elevated CEA levels pointing to benign liver or gastrointestinal diseases. Furthermore, we often rely on thresholds developed for other cancers to assess STM levels, which potentially limit their efficacy for MIBC. Another significant limitation stems from genetic variability, which can affect STM levels. A classic example is patients who lack Sialyl-Lewis (SL) genes genetically [30]. SL is the gene in charge of producing Sialyl Lewis antigen A (i.e., CA19-9). Approximately 10% of people are SL-negative globally, lacking the enzyme needed for CA 19-9 synthesis. In these SL-negative individuals, CA 19-9 remains undetectable, leading to false negatives, even in cases of BC. Thus, CA 19-9 is ineffective for this subset of patients.

## 6. Conclusions

In conclusion, despite these limitations affecting the diagnostic efficacy of STMs, they offer a cost-effective, non-invasive, and accessible tool for monitoring BC treatment response and detecting recurrence, provided the patient has an elevation in at least one STM prior to treatment. STMs hold particular promise in resource-limited settings where more advanced and costly diagnostic methods, such as ctDNA, may not be feasible. Future research should focus on improving the accuracy and clinical applicability of STMs, with the goal of fully integrating them into routine clinical practice.

## Figures and Tables

**Figure 1 cancers-17-00728-f001:**
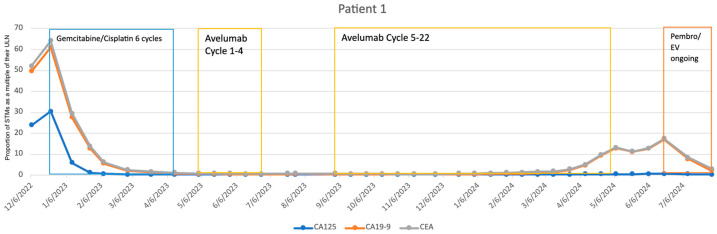
Patient 1’s treatment course with Serum Tumor Markers. Note: STM values have been adjusted to fit within a graphical space. STM values were adjusted by dividing by their values by their respective upper limit of normal (CA 19-9: 37 U/mL; CA-125: 35 U/mL; CEA: 3 U/mL). As a result, any value greater than 1 is representative of an abnormal lab finding.

**Figure 2 cancers-17-00728-f002:**
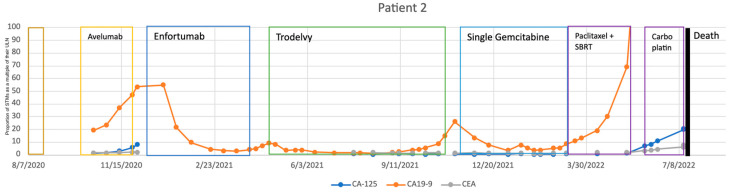
Patient 2’s treatment course with Serum Tumor Markers. Note: STM values have been adjusted to fit within a graphical space. STM values were adjusted by dividing by their values by their respective upper limit of normal (CA 19-9: 37 U/mL; CA-125: 35 U/mL; CEA: 3 U/mL). As a result, any value greater than 1 is representative of an abnormal lab finding.

**Figure 3 cancers-17-00728-f003:**
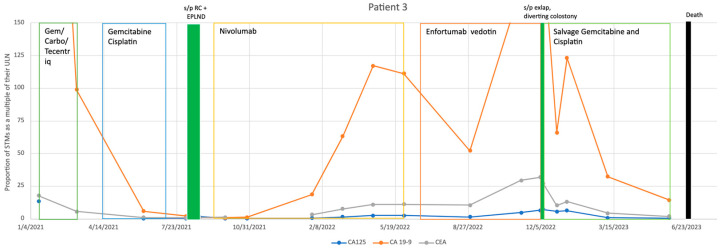
Patient 3’s treatment course with Serum Tumor Markers. Note: STM values have been adjusted to fit within a graphical space. STM values were adjusted by dividing by their values by their respective upper limit of normal (CA 19-9: 37 U/mL; CA-125: 35 U/mL; CEA: 3 U/mL). As a result, any value greater than 1 is representative of an abnormal lab finding.

## Data Availability

The data used to support this study are available from the corresponding author upon request.

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
