# Peer review of "Serum Tumor Markers for Muscle-Invasive Bladder Cancer in Clinical Practice: A Narrative Review"

_cancers, 2025, doi:10.3390/cancers17050728_

Round 1
Reviewer 1 Report
Comments and Suggestions for Authors
Dear Authors
Review article explains and concludes well that - Serum Tumor Markers (STMs) for Muscle-Invasive Bladder Cancer (MIBC) - the application of STMs —proteins in the blood that can signal the presence of cancer—as a diagnostic and monitoring tool for patients with MIBC, supplemented with real-life cases.
Minor corrections are required,
1) 2. STMs for Prognosis Assessment - Our institution was among the first to evaluate the potential role of STMs in BC starting in 2004. In the first study published from our institution in 2014, the prognostic value of CA-125 and CA 19-9 was evaluated. The results were promising, revealing a direct association between CA-125 levels and both extravesical extension and lymph node metastasis. Moreover, patients who had increased levels of CA-125 or CA 19-9 were shown to have worse overall survival [13]. Five years later and in our second study, we demonstrated that elevated precystectomy levels of CA 19-9 and CEA were independent predictors of worse 3-year overall survival, with 2.7- and 2-fold increased risk of death, respectively. Moreover, elevated CA 19-9 level was shown to be an independent predictor of an approximately 2.8-fold increase in recurrence risk at the 3-year follow up [4]. – Please keep correct reference numbers.
Author Response
Comment 1:Minor corrections are required, 1) 2. STMs for Prognosis Assessment - Our institution was among the first to evaluate the potential role of STMs in BC starting in 2004. In the first study published from our institution in 2014, the prognostic value of CA-125 and CA 19-9 was evaluated. The results were promising, revealing a direct association between CA-125 levels and both extravesical extension and lymph node metastasis. Moreover, patients who had increased levels of CA-125 or CA 19-9 were shown to have worse overall survival [13]. Five years later and in our second study, we demonstrated that elevated precystectomy levels of CA 19-9 and CEA were independent predictors of worse 3-year overall survival, with 2.7- and 2-fold increased risk of death, respectively. Moreover, elevated CA 19-9 level was shown to be an independent predictor of an approximately 2.8-fold increase in recurrence risk at the 3-year follow up [4]. – Please keep correct reference numbers.
Response 1: Thank you for your observation. We understand that the reference numbering may appear inconsistent. However, we would like to clarify that reference number 4, titled "Precystectomy Serum Levels of Carbohydrate Antigen 19-9, Carbohydrate Antigen 125, and Carcinoembryonic Antigen: Prognostic Value in Invasive Urothelial Carcinoma of the Bladder," was previously cited in the introduction. As a result, when it was cited again, the same reference number was used. Therefore, the numbering remains accurate despite appearing inconsistent.
Reviewer 2 Report
Comments and Suggestions for Authors
Doshi et al. have submitted an interesting narrative review manuscript with title “Serum Tumor Markers for Muscle-Invasive Bladder Cancer in Clinical Practice: A Narrative Review”.
Overall, the manuscript quality is high, text is well written, direct to the goal, and easy to follow. Figures are clear and the patients’ cases well explained. In addition, authors argument and discuss properly their findings, but also defend pretty decently the limitations of their study.
Considering the relevance of the topic, the quality of the study, I have no further corrections other than a minor comment.
Therefore, I suggest its acceptance for publication.
Minor comments:
Due to its relevance to the topic, I recommend to cite and comment the following references in the text.
Yuk HD, Han JH, Jeong SH, Jeong CW, Kwak C, Ku JH. Beta-human chorionic gonadotropin, carbohydrate antigen 19-9, cancer antigen 125, and carcinoembryonic antigen as prognostic and predictive biological markers in bladder cancer. Front Oncol. 2024 Dec 23;14:1479988. doi: 10.3389/fonc.2024.1479988. PMID: 39763612; PMCID: PMC11700811.
Stokkel LE, van Rossum HH, van de Kamp MW, Boellaard TN, Bekers EM, Kok NFM, van Rhijn BWG, Mertens LS. Clinical value of preoperative serum tumor markers CEA, CA19-9, CA125, and CA15-3 in surgically treated urachal cancer. Urol Oncol. 2023 Jul;41(7):326.e17-326.e24. doi: 10.1016/j.urolonc.2023.01.018. Epub 2023 Feb 21. PMID: 36813613.
Author Response
Comment 1:
Doshi et al. have submitted an interesting narrative review manuscript with title “Serum Tumor Markers for Muscle-Invasive Bladder Cancer in Clinical Practice: A Narrative Review”.
Overall, the manuscript quality is high, text is well written, direct to the goal, and easy to follow. Figures are clear and the patients’ cases well explained. In addition, authors argument and discuss properly their findings, but also defend pretty decently the limitations of their study.
Considering the relevance of the topic, the quality of the study, I have no further corrections other than a minor comment.
Therefore, I suggest its acceptance for publication.
Due to its relevance to the topic, I recommend to cite and comment the following references in the text.
Yuk HD, Han JH, Jeong SH, Jeong CW, Kwak C, Ku JH. Beta-human chorionic gonadotropin, carbohydrate antigen 19-9, cancer antigen 125, and carcinoembryonic antigen as prognostic and predictive biological markers in bladder cancer. Front Oncol. 2024 Dec 23;14:1479988. doi: 10.3389/fonc.2024.1479988. PMID: 39763612; PMCID: PMC11700811.
Stokkel LE, van Rossum HH, van de Kamp MW, Boellaard TN, Bekers EM, Kok NFM, van Rhijn BWG, Mertens LS. Clinical value of preoperative serum tumor markers CEA, CA19-9, CA125, and CA15-3 in surgically treated urachal cancer. Urol Oncol. 2023 Jul;41(7):326.e17-326.e24. doi: 10.1016/j.urolonc.2023.01.018. Epub 2023 Feb 21. PMID: 36813613.
Response 1:
Thank you for your comments and kind words. After reviewing the proposed manuscripts, we decided to include Yuk et. al to our paper as it helps strengthen our argument that certain pre-operative markers remain strong independent prognostic factors of overall survival. Please also review the revised manuscript, with changes highlighted. Since the other paper pertained to urachal cancer, we believed this paper would not be the best fit for our review.
Reviewer 3 Report
Comments and Suggestions for Authors
Review of the paper entitled “Serum Tumor Markers for Muscle-Invasive Bladder Cancer in Clinical Practice: A Narrative Review” by Chirag Doshi , Mazyar Zahir, Anosh Dadabhoy, Domenique Escobar, Leilei Xia and Siamak Daneshmand
The authors attempted to assess whether determining the level of serum tumor markers (STMs) such as Carbohydrate Antigen 19-9 (CA 19-9), Cancer Antigen-125 (CA-125) and carcinoembryonic antigen (CEA) in muscle-invasive bladder cancer (MIBC) patients can serve as effective prognostic tools. The topic taken up by the authors is interesting.
My comments
Despite the limitations affecting the diagnostic utility of the proposed STMs, the authors concluded that they offer a cost-effective, noninvasive, and accessible tool for monitoring the response to bladder cancer (BC) treatment and detecting relapse, provided that the patient has an elevated level of at least one STM before treatment.
Currently, many authors point out that a soluble fragment of cytokeratin 19, referred to as CYFRA 21-1, seems to be a promising marker of BC. Although the results of studies on the use of serum CYFRA 21-1 levels in the diagnosis and monitoring of BC are inconsistent, urinary CYFRA 21-1 levels seem to be extremely helpful in assessing the efficacy of treatment and in earlier identification of relapse. Literature date indicated that urine CYFRA 2 l-l levels were strongly correlated with tumor volume and were significantly better than urine cytology for the detection of bladder cancers of grades 1 and 2. It is strongly suggested that urine CYFRA 21-1 originates in tumor itself.
I would like the authors to provide their opinion and briefly discuss this issue in their manuscript. It may be worth considering using CYFRA 21-1 in combination with another tumor marker – such as CA 19-9, CA-125 or CEA. In my opinion, there is a good chance that the combined determination of CYFRA 21-1 in urine with CA 19-9, CA-125 or CEA in serum will increase the sensitivity of treatment monitoring.
Author Response
Comment 1:
Despite the limitations affecting the diagnostic utility of the proposed STMs, the authors concluded that they offer a cost-effective, noninvasive, and accessible tool for monitoring the response to bladder cancer (BC) treatment and detecting relapse, provided that the patient has an elevated level of at least one STM before treatment.
Currently, many authors point out that a soluble fragment of cytokeratin 19, referred to as CYFRA 21-1, seems to be a promising marker of BC. Although the results of studies on the use of serum CYFRA 21-1 levels in the diagnosis and monitoring of BC are inconsistent, urinary CYFRA 21-1 levels seem to be extremely helpful in assessing the efficacy of treatment and in earlier identification of relapse. Literature date indicated that urine CYFRA 2 l-l levels were strongly correlated with tumor volume and were significantly better than urine cytology for the detection of bladder cancers of grades 1 and 2. It is strongly suggested that urine CYFRA 21-1 originates in tumor itself.
I would like the authors to provide their opinion and briefly discuss this issue in their manuscript. It may be worth considering using CYFRA 21-1 in combination with another tumor marker – such as CA 19-9, CA-125 or CEA. In my opinion, there is a good chance that the combined determination of CYFRA 21-1 in urine with CA 19-9, CA-125 or CEA in serum will increase the sensitivity of treatment monitoring.
Response 1:
Thank you for your comment. After reviewing the literature, we agree that CYFRA 21-1 seems to be an exciting marker. However, CYFRA 21-1 does seem slightly out of scope for our paper, as we aimed to particularly review serum tumor markers that may be useful in the management and treatment response to muscle-invasive bladder cancer. Since most patients receive a radical cystectomy for MIBC, urinary markers are not as strong of a measure as compared to serum markers.
Reviewer 4 Report
Comments and Suggestions for Authors
This is a hybrid-format manuscript consisting of a review section and the presentation of three case reports that demonstrate the usability of serum tumor markers, CA 19-9, CA-125, and CEA, in bladder cancer. The authors discuss the application of these serum tumor markers for prognosis assessment, therapy response evaluation, and present their personal experience, which is illustrated through three clinical cases. The three cases are presented in the body of the manuscript and more detailed case presentation can be found in the Appendix A. There are three figures, one for each patient. The manuscript cites 21 references, which is relatively small but, considering the topic and content of the manuscript, can be considered adequate.
There are several previous reports from the same institution exploring serum tumor markers in bladder cancer. The authors and their team are pioneers in the field.
Although the use of serum tumor markers in bladder cancer has limited applicability in general clinical practice, this manuscript provides valuable insights into the potential role of CA 19-9, CA-125, and CEA in the management of bladder cancer. The manuscript can be accepted in its present form.
Author Response
Comment 1:
This is a hybrid-format manuscript consisting of a review section and the presentation of three case reports that demonstrate the usability of serum tumor markers, CA 19-9, CA-125, and CEA, in bladder cancer. The authors discuss the application of these serum tumor markers for prognosis assessment, therapy response evaluation, and present their personal experience, which is illustrated through three clinical cases. The three cases are presented in the body of the manuscript and more detailed case presentation can be found in the Appendix A. There are three figures, one for each patient. The manuscript cites 21 references, which is relatively small but, considering the topic and content of the manuscript, can be considered adequate.
There are several previous reports from the same institution exploring serum tumor markers in bladder cancer. The authors and their team are pioneers in the field.
Although the use of serum tumor markers in bladder cancer has limited applicability in general clinical practice, this manuscript provides valuable insights into the potential role of CA 19-9, CA-125, and CEA in the management of bladder cancer. The manuscript can be accepted in its present form.
Response 1:
Thank you so much for your great comments. We appreciate your highlights on the topic and made sure to include other publications that reference our insights. Please review our revised manuscript, with new text highlighted.